# Pain Perception during Orthodontic Treatment with Fixed Appliances

**Cristian Doru Olteanu [1], Sorana-Maria Bucur [2],*, Manuela Chibelean [3], Eugen Silviu Bud [3],*, Mariana Păcurar [3],* and Dana Gabriela Feștilă [1]**

[1] Orthodontic Department, Faculty of Dental Medicine, Iuliu Hațieganu University of Medicine and Pharmacy, 8 Babeș Str., 400012 Cluj-Napoca, Romania; cristidolteanu@yahoo.com (C.D.O.); dana.festila@umfcluj.ro (D.G.F.)

[2] Faculty of Medicine, Dimitrie Cantemir University of Târgu Mureș, 3-5 Bodoni Sandor Str., 540545 Târgu Mureș, Romania

[3] Orthodontic Department, Faculty of Dentistry, George Emil Palade University of Medicine, Pharmacy, Science and Technology, 540142 Târgu Mureș, Romania; manuela.chibelean@umfst.ro

* Correspondence: bucursoranamaria@gmail.com (S.-M.B.); eugen.bud@umfst.ro (E.S.B.); mariana.pacurar@umfst.ro (M.P.)

**Abstract:** The present study aimed to determine the intensity of pain perception in patients undergoing fixed orthodontic treatment. We analyzed the severity of pain concerning four routine procedures: the placement of separating elastics, ring cementations, arch activations, and elastic tractions. Our study consisted of a sample of 100 patients between 12 and 35 years old during the initial months of orthodontic treatment with fixed appliances. The patients completed a questionnaire meant to assess their pain sensation perception. The study sample was divided according to age and sex. By determining the intensity of pain felt during the four orthodontic procedures, we found that the most painful one was the ring cementation in all four age groups. The therapeutic-arch-activation procedure ranked second, with a higher mean value (2.66) in the 18–24 age group; the least painful was considered the elastic traction procedure, with a higher value (1.33) in the group over 30 years old. The most painful period was during the first 3–4 days after procedures. Most patients showed moderate pain after following the studied orthodontic interventions and required analgesic medication, the most frequently used being Nurofen, ketonal or paracetamol. The level of pain felt was significantly higher in men than in women. Patients suffer differently from the intensity of perceived pain as they grow older.

**Keywords:** pain perception; fixed appliances; orthodontic treatment

## 1. Introduction

The high incidence of dento-maxillary anomalies and patients' desire to improve facial appearance and esthetics have increased the number of patients seeking orthodontic treatment, especially with fixed appliances. Recent studies have shown that people, especially adolescents and adults, prefer to undergo a treatment with clear aligners to safeguard esthetics. Unfortunately, only a small number of them can afford the costs of this more comfortable and esthetic treatment type [1,2].

Pain sensation and masticatory discomfort are the most common side effects of orthodontic treatment. For efficient patient compliance and favorable outcomes, the orthodontist should inform patients about these aspects before starting treatment and should evaluate their motivation and potential risks. Moreover, it is ethically important to inform the patient about potential side effects of treatment, and it should be part of the patient's informed consent [3]. Orthodontic tooth displacement induces specific inflammatory reactions of the periodontium and the dental pulp that stimulate the release of various biochemical mediators, thus causing the sensation of pain [4,5]. Generally, the

action of orthodontic appliances consists of a change in the balance of the preexisting dentofacial forces, by triggering new ones or by favoring and selectively directing the action of natural forces.

Orthodontic forces depend on the properties of the appliances' materials (elasticity of wires/rubber rings, etc.), the construction particularities, and its interrelation with the dento-maxillary system [6–8]. The essential coordinates in orthodontic biomechanics are the orthodontic force, the weight-bearing area, and the application area, forming De *Nèvrezé*'s triad [9]. Force applied to the tooth crown is transmitted through the root to the periodontal ligament and the alveolar bone. For tooth displacement, alveolar bone resorption should occur in response to this stress; for the tooth to remain firmly attached, bone deposits should also develop to maintain the integrity of the attachment mechanism [10–13].

The International Association for the Study of Pain (IASP) defines pain as an unpleasant sensory and emotional experience associated with actual or potential tissue damage [14]. On the other hand, psychogenic pain is the pain generated by pure concurrent psychological factors strictly related to the person concerned or the environment in which that person lives. Psychogenic pain determined by the patient's psycho-emotional condition does not have a somatic origin [15]. Since pain is a subjective experience, we cannot evaluate it by indirect methods. In the case of children who cannot provide conclusive answers, the following parameters might be assessed:

- Physiological parameters: pulse, blood pressure, and sweating;
- Behavioral parameters: facial expressions (closed eyes, open mouth, dilated nostrils, hollow tongue), high-pitched crying, and body movements (kicking, closed fists) [16]. Still, there are some scales to determine pain intensity.
  1. Verbal scales—used at the time of presentation or during treatment, which classifies pain in 3, 5 or 7 grades: absent/weak/mild/moderate/intense/severe/extreme pain [15].
  2. Visual analog scales—these consist of a line with a length of 10 cm, oriented vertically/horizontally, whose extremes are two terms: "no pain" and "the greatest possible pain" [17].
  3. Numerical scales—in these cases, the patient is asked to quantify pain using numerical values between 0 and 10 or 0 and 100 (0 representing no pain and 10/100 extreme pain). These scales are easier to understand, providing at the same time a better description of pain intensity, and frequently replace verbal/visual scales [17].
  4. Behaviorally anchored scales—these quantify the intensity of pain based on its effect on behavior and describe the impact of pain on daily activities [17].

The study aimed to evaluate the intensity of pain perception in patients undergoing fixed orthodontic treatment. We analyzed the severity of pain concerning four routine procedures: the placement of separating elastics, ring cementation, arch ligation, and elastic traction. We wanted to determine which of these fixed orthodontic procedures causes pain to the patient, what is the pain intensity, and how long after performing the procedure does the intensity of the pain reach a maximum in order to be able to prepare them mentally for the appearance of pain and to indicate the appropriate medication. Determining which was the most used medication in fighting pain was also an objective of the study.

## 2. Materials and Methods

The current study was conducted at "Algocalm" Clinic of Târgu Mureș, Romania. A sample of 100 patients, children and adults aged 12–35 years with a good health status, were evaluated during the initial months of orthodontic treatment with fixed appliances. They were asked to complete a questionnaire to assess pain sensation perception. Thus, the patients answered what best fitted their perception of pain; using the numerical evaluation scale of our multiple-choice questionnaire (Appendix A), they could accurately assess the intensity level of pain [18]. Other pain implications, such as which was the most painful period after the procedure, when was the pain more intense, which medication was

more effective in fighting pain, and which was the most painful orthodontic procedure, were studied.

Informed consent was obtained from all the adult subjects involved in the study. For minor patients, informed consent was obtained from one of the parents. All the tested parameters were applied to both children and adults.

For the statistical analysis and the interpretation of the results, Microsoft Excel 2010 and GNU PSPP (Program for Statistical Analysis of Sampled Data) software programs version 1.5.3 (https://www.gnu.org/software/pspp/) were used, and the statistical *p*-value (representing the probability of a statistical test, being the lowest value of the significance level α for which the information extracted from the sample was significant) was determined using the Student *t*-test as well as the ANOVA results.

### 3. Results

Our sample included 100 patients divided into four age groups: 12–18 years (22%), 18–24 years (29%), 24–30 years (32%), and over 30 years (17%).

Distribution of pain intensity depending on the number of patients: the Numerical Pain Rating Scale ranged from 0 to 10 (0 being no pain, and 10 being extreme pain). Figure 1 shows the distribution of patients according to the age limits we have chosen. Figure 2 shows that of all 100 subjects included in the study, only 1 patient chose 2 as the lowest value, most of the patients chose the values 5 and 6 for pain intensity, and only 4 patients chose the highest value, 10. Following the calculations performed, the total mean of pain for the entire sample was 5.76.

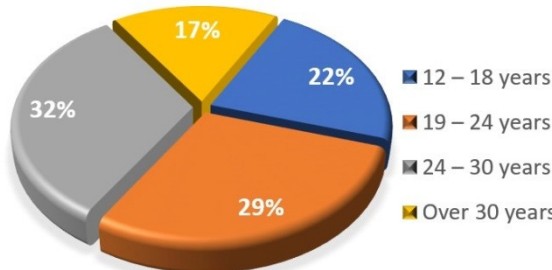

**Figure 1.** Distribution of patients by age groups.

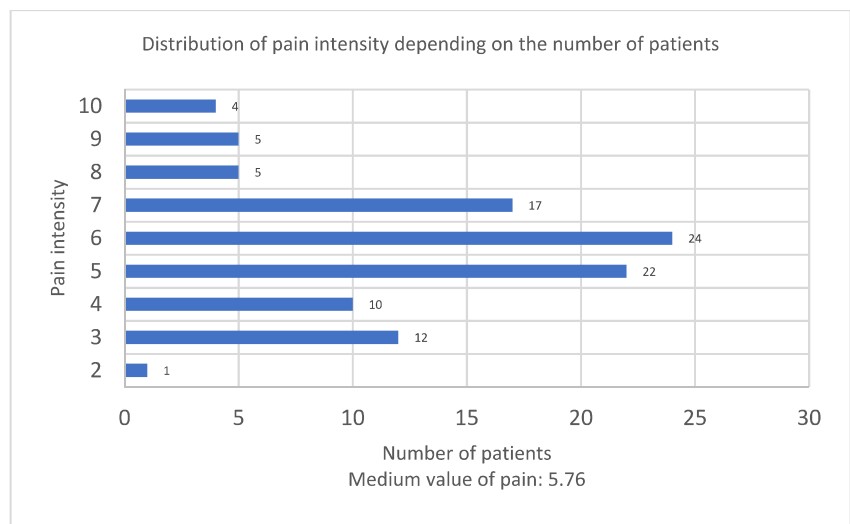

**Figure 2.** Distribution of pain intensity depending on the number of patients.

Distribution of patients depending on the period perceived as most painful: in the following diagram (Figure 3), the most painful period perceived by the subjects included in the study is analyzed. A significant number of them (29) chose the first 24 h as the most painful period, while the majority (40) selected the first 3–4 days as the most painful ones.

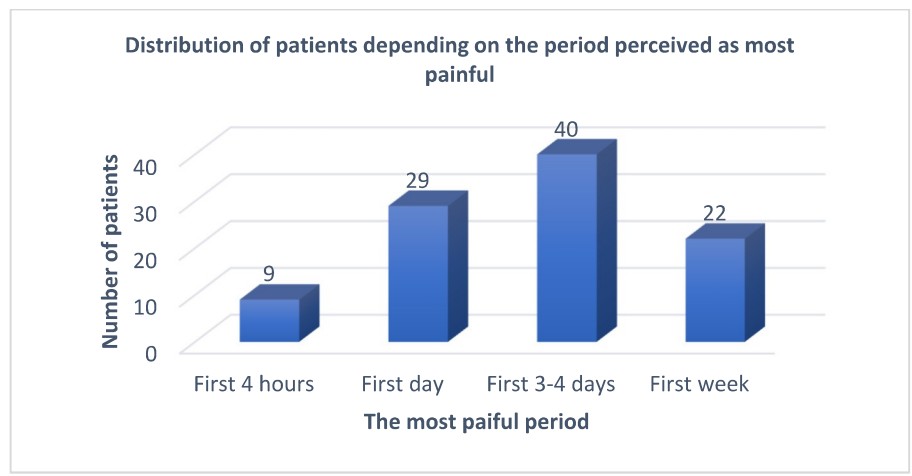

**Figure 3.** Distribution of patients depending on the period perceived as most painful.

Figure 4 shows that the perceived pain was significantly higher for the patients who needed pre-medication before interventions than for those who took no medication. At the same time, we may see that the mean pain value in men was higher than in women in both situations (with or without pre-medication).

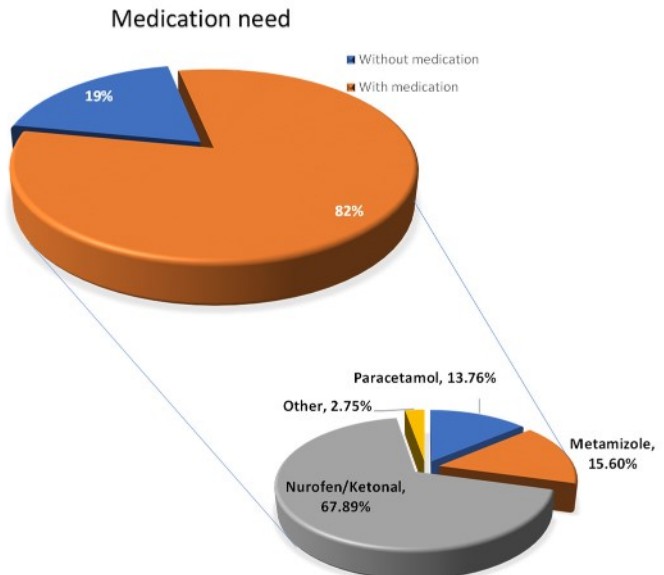

**Figure 4.** Distribution of patients depending on the type and the need for medication.

The perception of pain is different in our group depending on age, as is indicated by the Student $t$-test value obtained: t = 1.94, $p < 0.01$. We admit that people suffer the intensity of perceived pain differently as they grow older.

Concerning the need for medication after placement of orthodontic appliances, we can see that only 19% of the entire studied sample did not consider self-medication as necessary, whilst the vast majority, of 81%, chose to take medication to relieve the sensation of pain. Among those who used self-medication, almost 68% selected Ibuprofen, 15.6% chose Metamizole, 13.7% Paracetamol (Acetaminophen), and about 3% chose other drugs such as Codeine.

Figure 5 shows that for the females in the studied sample, the mean pain value was significantly lower than the mean value for the males.

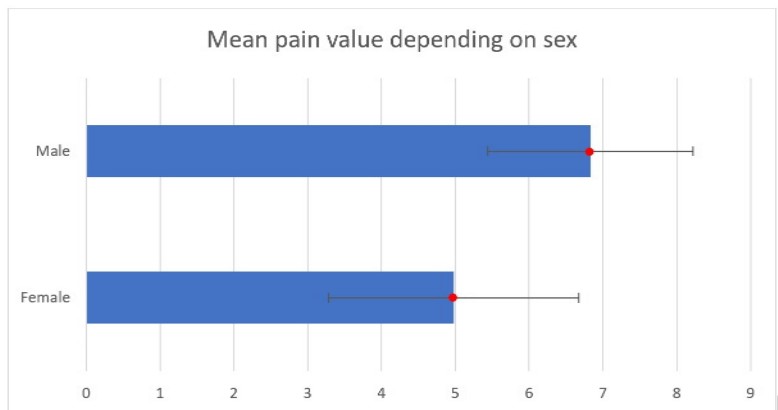

**Figure 5.** Graphical representation of pain value depending on sex.

There are differences in pain perception depending on gender, as indicated by the ANOVA results: F = 33.6, $p < 0.01$. Males suffered more than women from orthodontic procedures (interventions).

The diagram below (Figure 6) shows that the mean pain value varied during both daytime and night-time throughout the first 24 h.

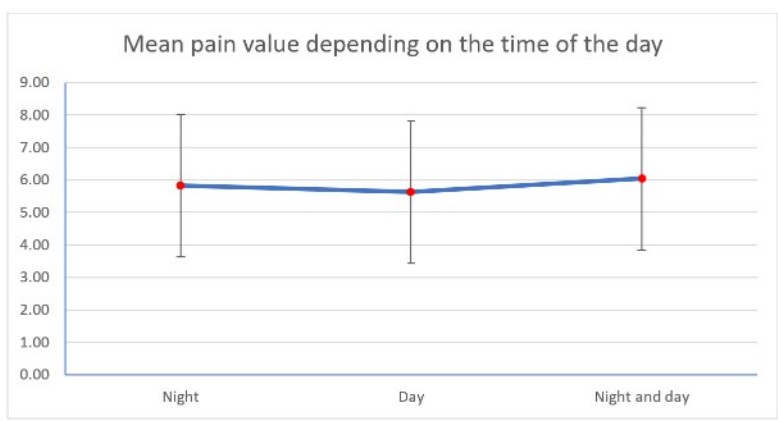

**Figure 6.** Graphical representation of pain value depending on the time of the day.

By analyzing the intensity of pain during the four most frequent orthodontic procedures, we may see that the most painful one was ring cementation in all four age groups, with a higher value for the group aged 18–24 years old (m = 2.83). The therapeutic-arch-activation procedure ranked second, with a higher mean value of 2.66 in the 18–24 age group, and the least painful was considered the elastic traction procedure, with a higher value of 1.33 in the group over 30 years old (Table 1, Figure 7).

**Table 1.** Pain intensity during the orthodontic procedures depending on age.

| Procedure | Age Group | Mean | *p* Value |
| --- | --- | --- | --- |
| Elastic separation | 12–18 | 2.09 | 0.04 |
| Ring cementation | 18–24 | 2.21 | 0.00 |
| Arch activation | 24–30 | 2.14 | 0.00 |
| Elastic traction | >30 | 1.07 | 0.04 |

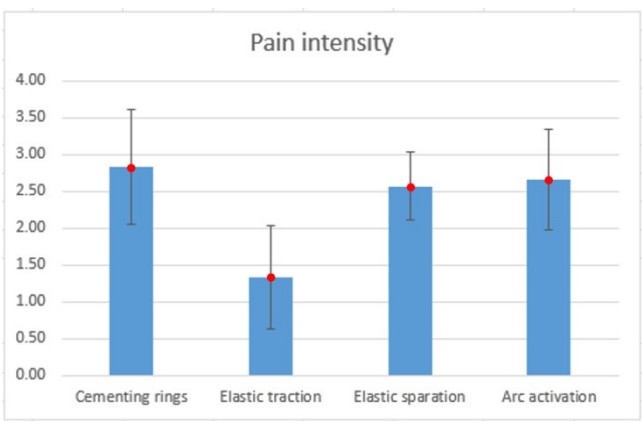

**Figure 7.** Graphical representation of pain intensity/procedure.

## 4. Discussion

The evaluation of pain is relevant for orthodontic treatments. Pain management supposes using specific verbal and non-verbal techniques to calm the patient and help them cope better with the pain [19]. Remarkable progress in pain study and mechanisms has been made over the past years. The visual analog scale, VAS, assesses the intensity of sensory pain and demonstrates that a non-invasive evaluation method is easy to use. Its disadvantage is the one-dimensional analysis of pain that is very complex and has many more particularities, even in usual social activities such as playing sports or meeting friends, attending school, etc. Interpersonal relations represent one of the most studied psychology issues. Social psychology concerns the relations between an individual and others [11].

Orthodontic forces should be applied 24 h/day to produce the most effective tooth displacement rate [20]. Evidence-based findings suggest that an orthodontic force action of at least 6 h/day is required to achieve minimal displacement. An optimal tooth displacement coefficient is considered to be of 1 mm/month. This movement may depend on the following factors:

- The force applied—both small and large forces cause tooth displacement, but in the case of smaller intensity forces that reduce and avoid periodontal ligament hyalinization, the displacement coefficient is bigger;
- The patient's age can influence the displacement coefficient; thus, in the case of adults, the tooth displacement rate is lower than in children due to the increased alveolar bone density and the diminution of cell response effectiveness [12].

We chose the age limit of 12, when permanent teeth are present in the oral cavity, and the correction of spacing/crowding is much easier. After 12 years, tooth displacement is assisted by facial development, and complete orthodontic treatment can be performed within a reasonable period of 18–24 months if the patient is compliant [21].

It is the age when behavioral disorders start to develop through self-neglect or, on the contrary, through a significant increase in interest in physical appearance, thus requiring a correction of dento-maxillary anomalies by orthodontic treatments, mainly for aesthetic considerations [22,23]. Following the calculations performed, the total mean pain value for the entire sample was 5.76. Specialized studies have reported that 8% of the studied samples, or about every 10th patient, discontinue orthodontic treatment because of the pain felt during the initial stages [24,25].

Regarding the period perceived as most painful, scientific studies mention that, generally, pain intensity increases over time from 4 h to 24 h after orthodontic procedures but decreases to normal after the first week [25–27]. Other studies report that pain perception starts several hours after putting on the orthodontic forces (with arch ligation to the brackets) and lasts about five days [28,29].

Most of the patients experienced moderate pain, with a pain level ranging between 5 and 7, and a mean value of 5.76.

After the placement of orthodontic appliances, if the patients considered medication as necessary, they were informed to select their own ones, excluding aspirin and Algocalmin, which have been shown to inhibit tooth movement, as proven in other studies [30]. The patients selected their medication based on previous experiences regarding pain release. According to other researchers, on the first days after each orthodontic appointment, patients take analgesics for pain caused by the dental appliances used for tooth displacement [21,31].

Regarding the mean pain value depending on age groups, the Student test performed showed that the $p$-value = 0.02642366, which is lower than the threshold $\alpha = 0.05$; hence, $p$ is statistically significant, and we may say that the pain perception depends on age.

Our results demonstrated that the mean pain value was different between the two sexes, regardless of age groups.

Studies show that the effect of age on the perception of the sensation of pain during orthodontic treatment is difficult to compare. Most authors suggest that adult subjects perceive a higher pain level than young subjects [26,27]. The 18–24 age group is usually associated with a higher degree of emotional disorders, which might explain the higher level of pain perception in this age group [9].

Some studies indicate that the male sex tolerates pain easily, considering the female sex more fragile and sensitive to the perception of the pain sensation caused by fixed orthodontic appliances [29,31]; other studies show no statistically significant difference between the two sexes [21].

Other authors such as Nandi et al. [32] showed that women more frequently present postoperative pain. Different explanations have been proposed for this: one of them [26] is based on the biological differences between the sexes. These refer to differences in the reproductive organs, which involve hormonal fluctuations associated with changes in serotonin and noradrenaline and an increase in the prevalence of pain during the menstrual cycle [21]. Ethical approaches interfere with pain perception in some cases [3]. On the other hand, one study [26] shows that the mean pain values were generally higher during the day than during the night, and reached a peak only during the first night, causing insomnia in the case of some patients (18%).

## 5. Conclusions

Most patients showed moderate pain following the studied orthodontic procedures.

1. The period perceived as most painful by patients treated with fixed orthodontic appliances was during the first 3–4 days after placement.
2. Most patients required analgesic medication during the fixed orthodontic treatment, the most frequent being Nurofen, ketonal, and paracetamol.
3. The mean value of the pain sensation felt was significantly higher in the case of the male sex compared to the female sex in all four age groups. Patients suffer the intensity of perceived pain differently as they grow older.
4. The most painful procedure was represented by ring cementation in all age groups, with the highest pain intensity in the 18–24 age group.
5. In cases where the clinical situation and the treatment plan allow, the cemented rings should be replaced with orthodontic tubes. If not, we will have to recommend an analgesic medication, especially 3 to 4 days after putting in the fixed orthodontic appliance.

**Author Contributions:** Conceptualization, C.D.O. and S.-M.B.; methodology, S.-M.B.; software, E.S.B.; validation, M.P., S.-M.B and D.G.F.; formal analysis, M.C.; investigation, C.D.O., E.S.B. and D.G.F.; resources, M.P.; data curation, M.C. and E.S.B.; writing—original draft preparation, S.-M.B.; writing—review & editing, S.-M.B.; visualization, M.C.; supervision, D.G.F. All authors have read and agreed to the published version of the manuscript.

**Funding:** This research received no external funding.

**Institutional Review Board Statement:** The study was conducted according to the guidelines of the Declaration of Helsinki, and approved by the Ethics Committee of SC Algocalm SRL, Targu-Mures, Romania, 917/06.05.2021.

**Informed Consent Statement:** Informed consent was obtained from all subjects involved in the study. For minor patients, informed consent was obtained from one of the parents.

**Data Availability Statement:** Data supporting reported results can be found by contacting Cristian Doru Olteanu, cristidolteanu@yahoo.com.

**Conflicts of Interest:** The authors declare no conflict of interest.

## Appendix A

*Questionnaire*

Evaluation of the perception of the pain sensation during the initial stages of orthodontic treatment with fixed appliances. Personal data will be used strictly for academic purposes, while respecting confidentiality!

1.   Name:
2.   Age:

    (a)   12 to 18 years
    (b)   18 to 24 years
    (c)   24 to 30 years
    (d)   over 30 years

3.   Sex:

    (a)   male
    (b)   female
    (c)   other

4.   General health status:

    (a)   Good

Affected (mention the disease, e.g., allergies, heart diseases, diabetes, bleeding disorders, leukemia, juvenile rheumatoid arthritis, renal failure, endocarditis, cystic fibrosis).

5.   The most painful period was:

    (a)   the first 4 h
    (b)   the first day
    (c)   the first 3–4 days
    (d)   the first week

6.   Pain was more intense:

    (a)   during the day
    (b)   at night
    (c)   during the day and the night

7.   Mention the level of the perceived pain on a numerical scale from 0 to 10 (0 being no pain and 10 being extreme pain):

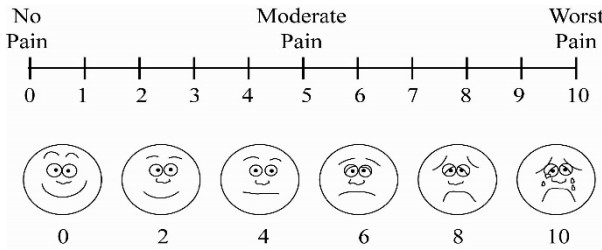

**Figure A1.** Visual representation of pain perception.

8.  The drugs administered for pain relief were: Which was the most effective in fighting pain

    (a)  Ketonal/Ibuprofen/Nurofen
    (b)  Metamizole
    (c)  Paracetamol
    (d)  Other (mention them)

9.  Mention the most painful procedure:

    (a)  ring cementation
    (b)  use of separating elastics
    (c)  arch activation
    (d)  elastic traction

    Date, Signature

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
