# Peer review of "Pain Perception during Orthodontic Treatment with Fixed Appliances"

_applsci, doi:10.3390/app12136389_

Round 1

Reviewer 1 Report

The topic of this article is not original. Many studies have already evaluated the pain perception during the orthodontic treatment. However, some suggestions may be considered as follows:

Abstract

-       The abstract is too short. It should also be more informative and specific.

-       Please, add more information about the search strategy.

-       Please, define the aims of this study clearly.

Keywords

-       You should modify the keywords “pain” and “orthodontics” in “pain perception” and “orthodontic treatment”, respectively.

Introduction

-       The introduction should be corrected, especially the first sentence
“The high frequency of dento-maxillary anomalies and the wish to improve facial appearance have caused an increase in patients’ attendance to orthodontic services for orthodontic treatment, mainly with fixed appliances”. Recent studies have shown that people, especially adolescents and adults, prefer to undergo a treatment with clear aligners to safeguard aesthetics.
In addition, some references should be included.

-       You should add some references at the end of this sentence “Orthodontic tooth displacement induces specific inflammatory reactions of the periodontium and the dental pulp that stimulate the release of various biochemical mediators, thus causing the sensation of pain”. For example, the following reference should be added to improve the discussion of the manuscript:

1.     d'Apuzzo F, Nucci L, Delfino I, Portaccio M, Minervini G, Isola G, Serino I, Camerlingo C, Lepore M. Application of Vibrational Spectroscopies in the Qualitative Analysis of Gingival Crevicular Fluid and Periodontal Ligament during Orthodontic Tooth Movement. J Clin Med 2021;10(7):1405. doi: 10.3390/jcm10071405

-       You should add more references at the end of the sentence “Orthodontic forces depend on the properties of the appliances' materials (elasticity of wires/rubber rings, etc.), the construction particularities, and its interrelation with the dento-maxillary system [1]”.

-        “For the tooth displacement, alveolar bone resorption should occur in response to this stress; for the tooth to remain firmly attached, bone deposits should also develop to maintain the integrity of the attachment mechanism [3-6]” - references n. 4 and n.6 are not pertinent.

-       “The International Association for the Study of Pain (IASP) defines pain as an unpleasant sensory and emotional experience associated with actual or potential tissue damage [7]” – reference n. 7 is not pertinent. You should add some references about the IASP.

-       Psychogenic pain determined by the patient’s psycho-emotional condition does not have a somatic origin [8].” - references n. 8 is not pertinent. For example, the following reference should be added to improve the discussion of the manuscript:

1.     Rotolo RP, Nucci L, Grassia V, Perillo L, d’Apuzzo F. Bullying and malocclusion in adolescence: a case report. South Eur J Orthod Dentofacial Res 2020

-       Please, specify clearly the parameters assessed. Did only children use physiological parameters and behavioral parameters? Which parameters were used for adults? I suggest to insert these sentences in the paragraph “Materials and Methods”.

-       Please, specify crealrly the aim of the study. What is the real purpose of this study?

Materials and methods

-       Some parts of this section are written in the Introduction.

-       Did the children’s parents sign the informed consent?

Results

-       This section should be rewritten. There is a lot of confusion. Please, pay attention to the punctuation.

-       Please, define clearly the sentence “Mean pain value depending on age groups, by sex - the p result of the statistical test expressed as a whole number between 0 and 1 represents the probability of making an error if rejected the H0 hypothesis”.

Discussions

-       Some references should be added.

-       You should add some references at the end of this sentence “Ideally, orthodontic forces should be applied 24 hours/day to produce the most effective tooth displacement rate”.

-       Reference n.14 is not pertinent “Specialized studies have reported that 8% of the studied samples or about every 10th patient discontinues orthodontic treatment because of the pain felt during the initial stages [14-15].”

-       You should remove reference n.21 and add reference n. 14 at the end of the sentence “Among those who used self-medication, almost 68% selected Ibuprofen, 15.6% chose Metamizole, 13.7% Paracetamol (Acetaminophen), and about 3% chose other drugs like Codeine, Aspirin. According to other researchers on the first day after each orthodontic appointment, patients take analgesics for pain caused by the dental appliances used for tooth displacement [20-21].”

References

-       The articles should be numbered and cited in the text with related numbers.

-       Please, remove the references that are not inherent to the study.

Language

-       The English language should be improved; a native speaker should revise the manuscript before resubmission.

Figures and Tables

-       Figure 1 and 7 do not provide any others information compared to the text. They should be removed;

-       Table I is not simple to read, also it is not clearly explained in the text.

Author Response

Dear reviewer,

Thank you for the suggestions that helped us improve the article. Thank you very much for the time spent reading the paper, and give clear advice.

The topic of this article is not original. Many studies have already evaluated pain perception during orthodontic treatment.

It is a good point, but this topic has been not investigated before in our country. As practitioners, we are becoming more and more aware of the comfort of our patients, not only about orthodontic results. We thought the study would be useful in our activity.

Abstract

-       The abstract is too short. It should also be more informative and specific.

-       Please, add more information about the search strategy.

-       Please, define the aims of this study clearly.

We have changed the abstract according to your suggestions.

Keywords

-       You should modify the keywords “pain” and “orthodontics” in “pain perception” and “orthodontic treatment”, respectively.

We have done the changes.

Introduction

-       The introduction should be corrected, especially the first sentence
“The high frequency of dento-maxillary anomalies and the wish to improve facial appearance have caused an increase in patients’ attendance to orthodontic services for orthodontic treatment, mainly with fixed appliances”. Recent studies have shown that people, especially adolescents and adults, prefer to undergo a treatment with clear aligners to safeguard aesthetics.
In addition, some references should be included.

We added comments and references. We totally agree with you regarding the patient’s preference for clear aligners but only a small part of them can afford the costs of this more comfortable and esthetic treatment type.

-       You should add some references at the end of this sentence “Orthodontic tooth displacement induces specific inflammatory reactions of the periodontium and the dental pulp that stimulate the release of various biochemical mediators, thus causing the sensation of pain”. For example, the following reference should be added to improve the discussion of the manuscript:

  1. d'Apuzzo F, Nucci L, Delfino I, Portaccio M, Minervini G, Isola G, Serino I, Camerlingo C, Lepore M. Application of Vibrational Spectroscopies in the Qualitative Analysis of Gingival Crevicular Fluid and Periodontal Ligament during Orthodontic Tooth Movement. J Clin Med 2021;10(7):1405. doi: 10.3390/jcm10071405

We have added. Excellent idea, thank you!

-       You should add more references at the end of the sentence “Orthodontic forces depend on the properties of the appliances' materials (elasticity of wires/rubber rings, etc.), the construction particularities, and its interrelation with the dento-maxillary system [1]”.

We have added:

Mansour AY. A comparison of orthodontic elastic forces: Focus on reduced inventory. J Orthod Sci. 2017;6(4):136-140. doi:10.4103/jos.JOS_58_17

Nucera R, Gatto E, Borsellino C, Aceto P, Fabiano F, Matarese G, Perillo L, Cordasco G. Influence of bracket-slot design on the forces released by superelastic nickel-titanium alignment wires in different deflection configurations. Angle Orthod. 2014 May;84(3):541-7. doi: 10.2319/060213-416.1.

-        “For the tooth displacement, alveolar bone resorption should occur in response to this stress; for the tooth to remain firmly attached, bone deposits should also develop to maintain the integrity of the attachment mechanism [3-6]” - references n. 4 and n.6 are not pertinent.

We have replaced the initial 2references 4 and 6 by:

d'Apuzzo F, Cappabianca S, Ciavarella D, Monsurrò A, Silvestrini-Biavati A, Perillo L, Biomarkers of Periodontal Tissue Remodeling during Orthodontic Tooth Movement in Mice and Men: Overview and Clinical Relevance. The Scientific World Journal, vol. 2013, Article ID 105873, 8 pages, 2013. https://doi.org/10.1155/2013/105873.

Respectively

Jeon HH, Teixeira H, Tsai A. Mechanistic Insight into Orthodontic Tooth Movement Based on Animal Studies: A Critical Review. J Clin Med. 2021;10(8):1733. Published 2021 Apr 16. doi:10.3390/jcm10081733.

-       “The International Association for the Study of Pain (IASP) defines pain as an unpleasant sensory and emotional experience associated with actual or potential tissue damage [7]” – reference n. 7 is not pertinent. You should add some references about the IASP.

We have replaced the reference by:

Raja SN, Carr DB, Cohen M, et al. The revised International Association for the Study of Pain definition of pain: concepts, challenges, and compromises. Pain. 2020;161(9):1976-1982. doi:10.1097/j.pain.0000000000001939 

-       “Psychogenic pain determined by the patient’s psycho-emotional condition does not have a somatic origin [8].” - reference n. 8 is not pertinent. For example, the following reference should be added to improve the discussion of the manuscript:

  1. Rotolo RP, Nucci L, Grassia V, Perillo L, d’Apuzzo F. Bullying and malocclusion in adolescence: a case report. South Eur J Orthod Dentofacial Res 2020

We have replaced. Excellent idea, thank you!

-       Please, specify clearly the parameters assessed. Did only children use physiological parameters and behavioral parameters? Which parameters were used for adults? I suggest inserting these sentences in the paragraph “Materials and Methods”.

We have specified the parameters assessed. All of the parameters were applied to children and adults, too.

-       Please, specify clearly the aim of the study. What is the real purpose of this study?

We wanted to determine which of these fixed orthodontic procedures causes pain to the patient, what is the pain intensity, and how long after performing the procedure the intensity of pain gets a maximum in order to be able to prepare him/her mentally for the appearance of pain and to indicate the appropriate medication.

Materials and methods

-       Some parts of this section are written in the Introduction.

-       Did the children’s parents sign the informed consent?

Informed consent was obtained from all the adult subjects involved in the study. For minor patients, informed consent was obtained from one of the parents.

Results

-       This section should be rewritten. There is a lot of confusion. Please, pay attention to the punctuation.

-       Please, define clearly the sentence “Mean pain value depending on age groups, by sex - the p result of the statistical test expressed as a whole number between 0 and 1 represents the probability of making an error if rejected the H0 hypothesis”.

We rewrote the results

Discussions

-       Some references should be added.

-       You should add some references at the end of this sentence “Ideally, orthodontic forces should be applied 24 hours/day to produce the most effective tooth displacement rate”.

Asiry MA. Biological aspects of orthodontic tooth movement: A review of literature. Saudi J Biol Sci. 2018;25(6):1027-1032. doi:10.1016/j.sjbs.2018.03.008

-       Reference n.14 is not pertinent “Specialized studies have reported that 8% of the studied samples or about every 10th patient discontinues orthodontic treatment because of the pain felt during the initial stages [14-15].”

Rakhshan H, Rakhshan V. Pain and discomfort perceived during the initial stage of active fixed orthodontic treatment. Saudi Dent J. 2015;27(2):81-87. doi:10.1016/j.sdentj.2014.11.002

-       You should remove reference n.21 and add reference n. 14 at the end of the sentence “Among those who used self-medication, almost 68% selected Ibuprofen, 15.6% chose Metamizole, 13.7% Paracetamol (Acetaminophen), and about 3% chose other drugs like Codeine, Aspirin. According to other researchers on the first day after each orthodontic appointment, patients take analgesics for pain caused by the dental appliances used for tooth displacement [20-21].”

Done

References

-       The articles should be numbered and cited in the text with related numbers.

-       Please, remove the references that are not inherent to the study.

Done

Language

-       The English language should be improved; a native speaker should revise the manuscript before resubmission.

The manuscript has been revised by an English native speaker.

Figures and Tables

- Figures 1 and 7 do not provide any other information than the text. They should be removed;

We decided to maintain these two figures together with text explications because we wanted to provide also visual images to the readers. We hope it was the right decision.

-       Table I is not simple to read, also it is not clearly explained in the text.

We have removed Tables I and II because at a closer look they were not necessary. We have textually presented the results. We hope that the other reviewers will like it. Table III was modified.

We all and I personally thank you again. If something is still wrong, please let us know. We really want the article to be published.

My best regards,

Sorana-Maria Bucur

Reviewer 2 Report

I express my deep appreciation and respect for the work done by the authors of this article. This paper presents interesting and original results of experimental studies on pain perception depending on the types of procedures of orthodontic treatment. I read the manuscript quickly and easily, all the results are presented clearly. The results in the article are presented in such a way that the article can be understood by a wide range of people, not only specialists in the field of surgical dentistry. The significance level/threshold of 5% was chosen within reasonable limits, which ensures the acceptable reliability of the results and conclusions. However, I would like to outline a few comments and questions that I hope will help improve the quality of this work. The comments are listed below.

1.    Abstract: not structured properly. The method, materials and main conclusions of the study should be added to the «Abstract» section.

2.    The narrative line of the introduction is quite clear and logical, but there is a obviously lack of literary references.

3.    There is no logical conclusion in the «Introduction» of what for where further research will be directed. In other words, what practical application or clinical actions will be taken based on the results obtained in this article.

4.    Authors should check the correctness of following the template according to the requirements of the journal, spelling, punctuation and subscripts. Alter the drawings so that the image is better and more clearly visible.

5.    The Student's criterion is used as a statistical method. However, is it not necessary to apply the Fisher criterion with the same value of significance levels (5%) to assess the homogeneity (comparable dispersion) of all samples (sex/gender, age restrictions, treatment time)?

6.    Please emphasize in the «Conclusion» section the importance of the results you have obtained and their applicability, for example, for the correction of treatment, the search for new methods of implantation or the introduction of more inert and safe painkillers.

Author Response

Dear reviewer,

Thank you for the appreciation and the suggestions that helped us improve the article. Thank you very much for the time spent reading the paper, and give clear advice.

  1. Abstract: not structured properly. The method, materials and main conclusions of the study should be added to the «Abstract» section.

We have modified the abstract according to your suggestions.

  1. The narrative line of the introduction is quite clear and logical, but there is a obviously lack of literary references.

We have added references.

  1. There is no logical conclusion in the «Introduction» of what for where further research will be directed. In other words, what practical application or clinical actions will be taken based on the results obtained in this article.

We have explained the aim of the study.

  1. Authors should check the correctness of following the template according to the requirements of the journal, spelling, punctuation and subscripts. Alter the drawings so that the image is better and more clearly visible.
  2. The Student's criterion is used as a statistical method. However, is it not necessary to apply the Fisher criterion with the same value of significance levels (5%) to assess the homogeneity (comparable dispersion) of all samples (sex/gender, age restrictions, treatment time)?

We have decided not to use the Fisher test because the sample of the study is small, and it was divided anyway into small groups according to age, sex, etc.

  1. Please emphasize in the «Conclusion» section the importance of the results you have obtained and their applicability, for example, for the correction of treatment, the search for new methods of implantation or the introduction of more inert and safe painkillers.

We have explained that, and we have improved the conclusions.

We all and I personally thank you again. If something is still wrong, please let us know. We really want the article to be published.

My best regards,

Sorana-Maria Bucur

Reviewer 3 Report

The research paper will need major revisions, grammatical corrections and modification in the patient questionnaire. Please rephrase the paper and submit. 

The following lines need corrections-

10: you may use words like "determine" instead of investigate

14: initial months rather than "first months"

20: "were studied" instead of "was investigated"

26-29: Please rephrase- for e.g. you can write: The high incidence of dento-maxillary anomalies and patients' desire to improve facial appearance/esthetics have resulted in increase in number of patients seeking orthodontic treatment especially with fixed appliances". 

30: Effects instead of "effect"

30-32: Change wordings- for an efficient patient compliance and favorable outcomes, orthodontist should inform patients...Also, it is ethically important to inform patient about potential side effects of treatment and it should be part of patients' informed consent.

88: Were evaluated, instead of "was evaluated".

127-131: Please explain these lines in detail- self-medication and premedication are different 

175:176- Rephrase

183: playing sports instead of "doing sports"

188: Clinical experience is very subjective term. Evidence based findings are preferred.

Questionnaire related-

Age groups overlap. For e.g. 24 years seems to be part of both first and second age groups. you can modify as 12-17,18-24,25-30,31+...have to be specific in questionnaire (31st December of 2000 may be considered as end date for a certain age group and 1st January of 2001 as start date of another age group. This is just an example, patients need to identify their age group correctly without getting confused.

Sex: Male/Female and Other is recommended as there may be some patients who wish to not identify as male or female and may fall in other category. If necessary you can elaborate further

General Health status : Please describe more making it easier to understand for patients

The sample size of 100 is very small and can be increased.

Author Response

Dear reviewer,

Thank you for the appreciation and the suggestions that helped us improve the article. Thank you very much for the time spent reading the paper, and give clear advice.

The research paper will need major revisions, grammatical corrections and modification in the patient questionnaire. Please rephrase the paper and submit. 

We have done a lot of changes to the content and I have rephrased the paper. The paper was checked by an English native speaker.

The following lines need corrections-

10: you may use words like "determine" instead of investigate

We have rephrased.

14: initial months rather than "first months"

We have rephrased.

20: "were studied" instead of "was investigated"

We have rephrased.

26-29: Please rephrase- for e.g. you can write: The high incidence of dento-maxillary anomalies and patients' desire to improve facial appearance/esthetics have resulted in increase in number of patients seeking orthodontic treatment especially with fixed appliances". 

We have rephrased.

30: Effects instead of "effect"

Done.

30-32: Change wordings- for an efficient patient compliance and favorable outcomes, orthodontist should inform patients...Also, it is ethically important to inform patient about potential side effects of treatment and it should be part of patients' informed consent.

We have rephrased.

88: Were evaluated, instead of "was evaluated".

Done.

127-131: Please explain these lines in detail- self-medication and premedication are different 

Done.

175:176- Rephrase

Done

183: playing sports instead of "doing sports"

Changed

188: Clinical experience is very subjective term. Evidence based findings are preferred.

O.K.

Questionnaire related-

Age groups overlap. For e.g. 24 years seems to be part of both first and second age groups. you can modify as 12-17,18-24,25-30,31+...have to be specific in questionnaire (31st December of 2000 may be considered as end date for a certain age group and 1st January of 2001 as start date of another age group. This is just an example, patients need to identify their age group correctly without getting confused.

We have found a solution.

Sex: Male/Female and Other is recommended as there may be some patients who wish to not identify as male or female and may fall in other category. If necessary you can elaborate further

Done.

General Health status : Please describe more making it easier to understand for patients

The inspiration was

Donald Burden, Brian Mullally, Jonathan Sandler, Orthodontic treatment of patients with medical disorders, European Journal of Orthodontics, Volume 23, Issue 4, August 2001, Pages 363–372, https://doi.org/10.1093/ejo/23.4.363

The sample size of 100 is very small and can be increased.

We agree with you. The motivation for the small number of new patients was the pandemic which caused the decrease in the addressability of our orthodontic services. That is why the study took place in two different locations, in two cities.

We all and I personally thank you again. If something is still wrong, please let us know. We really want the article to be published.

My best regards,

Sorana-Maria Bucur

Reviewer 4 Report

melhorar o estilo do texto e as referências bibliográficas 

Author Response

Dear reviewer,

Thank you for the appreciation and the suggestions.

Melhorar o estilo do texto e as referências bibliográficas 

The paper was greatly improved in content and references.

We all and I personally thank you again. If something is still wrong, please let us know. We really want the article to be published.

My best regards,

Sorana-Maria Bucur

Round 2

Reviewer 1 Report

Dear Authors, 

The paper was correctly revised.
Thus, I think it may be now accepted. 

Best regards, 
Ludovica Nucci 

Author Response

Dear reviewer,

Thank you again for the help that you have given to improve the article.

The authors

Reviewer 3 Report

Thank you for responding to the feedback.

Author Response

We have improved the research design. Thank you very much!